# CL-NeRF: Continual Learning of Neural Radiance Fields for Evolving Scene Representation

**Xiuzhe Wu**$^\diamond$  **Peng Dai**$^{\diamond *}$  **Weipeng Deng**$^{\diamond *}$  **Handi Chen**$^\diamond$
**Yang Wu**$^\dagger$  **Yan-Pei Cao**$^\ddagger$  **Ying Shan**$^\ddagger$  **Xiaojuan Qi**$^\diamond$

$^\diamond$The University of Hong Kong
$^\dagger$Tencent AI Lab    $^\ddagger$ARC Lab, Tencent PCG

## Abstract

Existing methods for adapting Neural Radiance Fields (NeRFs) to scene changes require extensive data capture and model retraining, which is both time-consuming and labor-intensive. In this paper, we tackle the challenge of efficiently adapting NeRFs to real-world scene changes over time using a few new images while retaining the memory of unaltered areas, focusing on the continual learning aspect of NeRFs. To this end, we propose CL-NeRF, which consists of two key components: a lightweight expert adaptor for adapting to new changes and evolving scene representations and a conflict-aware knowledge distillation learning objective for memorizing unchanged parts. We also present a new benchmark for evaluating Continual Learning of NeRFs with comprehensive metrics. Our extensive experiments demonstrate that CL-NeRF can synthesize high-quality novel views of both changed and unchanged regions with high training efficiency, surpassing existing methods in terms of reducing forgetting and adapting to changes. Code and benchmark will be made available.

## 1   Introduction

Neural Radiance Fields (NeRFs), powerful models capable of synthesizing high-fidelity novel views from multi-view images [2, 26, 7], play crucial roles in various applications including AR/VR and autonomous driving simulators. However, real-world scenes constantly evolve over time, such as a chair being removed from a room. To handle such changes, it is essential to develop adaptable representations that can be continuously updated to reflect these changes accurately with minimal data and computational cost. However, existing NeRF methods [2, 26, 7, 43, 41, 40] often require a lot of new data and full retraining of the model to adapt to even minor local changes, which is impractical and costly, especially for large-scale scenes e.g., city-scale scene.

In this paper, we study a novel efficient continual learning task for NeRF, where we aim to update a scene's NeRF using only a few new images (i.e., 10-20 images) that capture the scene changes over time while preserving the memory of the unchanged areas (see Figure 1). A simple approach is to fine-tune the NeRF with the new images, but this suffers from catastrophic forgetting of the unaltered areas (see Figure 1). One might argue that using explicit 3D representations, [9, 22, 27], such as Plenoxels [9], could mitigate this problem since the explicit 3D representation might not change for the unchanged regions during fine-tuning. Nonetheless, our initial experiments suggest that forgetting still occurs (refer to Figure 1). This is because the ray-based volumetric rendering formula [2, 26, 9, 22, 27] propagates supervision signals to all voxels along a ray, which may affect other regions that share voxels with the same ray. The most related work is dynamic NeRFs [17, 16, 30], which also model dynamic changes using time-aware latent codes [17] or scene flows [19]. But it focuses on changes

---

$^*$Equal contribution.

37th Conference on Neural Information Processing Systems (NeurIPS 2023).

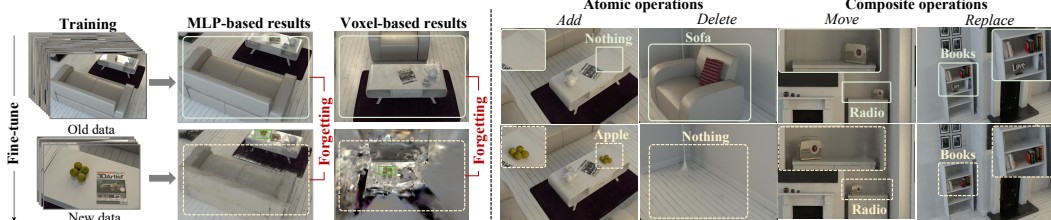

Figure 1: Catastrophic forgetting (left) and scene changes simulation (right). The left part illustrates the forgetting behavior of MLP-based and Voxel-based NeRF during fine-tuning, while the right side simulates representative real-world change operations.

caused by continuous object motions and is designed to remember the observed regions at a specific timestamp, leading to forgetting of unobserved regions in a large-scale scene.

To address this challenge, we introduce Continual Learning NeRF (CL-NeRF), which efficiently adapts a learned NeRF model to environmental changes using minimal new data while retaining its memory of unchanged areas. It consists of two core components: an expert adaptor for adapting to scene changes and a conflict-aware distillation scheme for memorizing unchanged parts. The expert adaptor is a lightweight network that learns to encode local changes from a few new images while keeping the original NeRF parameters fixed. It also predicts a weight value called mask logit that indicates the likelihood of the rendering point being in the changed areas. It is used to fuse the original NeRF features and the expert adaptor for rendering. Further, the conflict-aware knowledge distillation scheme trains the model to preserve the original scene representation for unchanged areas by minimizing the discrepancy between the outputs of the original NeRF and the adapted model. The mask logit predicted by the expert network balances the distillation loss and rendering loss of new images, avoiding conflicting supervision signals.

Finally, we introduce a new continual learning benchmark for NeRF, consisting of synthetic and real-world scenes. Our benchmark encompasses four fundamental operations that represent prevalent real-world scene changes: add, remove, move, and replace. Furthermore, to facilitate evaluation, we develop comprehensive metrics to assess the model's effectiveness in preserving unaltered areas and adapting to new changes in terms of accuracy and efficiency. Our extensive experiments, involving three baselines (i.e., fine-tuning, memory replay, and DyNeRF[17]), demonstrate that our model consistently outperforms all compared approaches on all evaluated scenarios while maintaining superior efficiency. Especially, our approach maintains high-quality results throughout a series of operations in both old and new tasks as evidenced by our PSNR scores (29.74 and 33.38 in the new and old task, respectively.) Notably, our approach effectively mitigates the forgetting phenomenon when compared to direct fine-tuning (see Figure 1).

In summary, our primary contributions are:

1) To our best knowledge, we present the first study on Continual Learning of NeRF, aiming to enable a NeRF model to adapt to changes in real-world scenes. We also introduce a novel benchmark for this task, which can serve as a valuable resource for future research.

2) We propose a new method – CL-NeRF – consisting of two key components: a lightweight expert adaptor for adapting to new changes and a conflict-aware knowledge-distillation objective for preserving unaltered scene representations.

3) We conduct extensive experiments and show that our model achieves high-quality view synthesis in both altered and unaltered regions, outperforming the baseline methods and mitigating the forgetting phenomenon.

## 2   Related Works

**Neural Radiance Fields, Dynamic NeRF and NeRF Editing**    Neural Radiance Fields (NeRF) [26] have drawn a lot of attention recently due to their high-fidelity rendering quality. To render novel-view images, previous methods encode the geometry and appearance into the weights of MLPs [26, 2, 3] or other explicit neural 3D representations [22, 9], then render novel-view images through volume

rendering [12] based on the predicted radiance values of query points. However, most of these methods require per-scene optimization and are fragile to the scene changes [35].

To handle scene changes, the vanilla NeRF [26] is modified to model the dynamic objects in the scene. This line of work usually takes videos containing dynamic objects as inputs and can be divided into two lines: deformation field estimation [19, 20, 28, 31, 39, 29, 8] and the incorporation of time-aware inputs [17, 4, 36]. Dycheck [10] offers a critical assessment of recent developments in dynamic NeRFs. Given that the dynamic NeRFs are only trained on limited observations of the large-scale scene at each timestamp, thus the high-fidelity rendering at that timestamp is constrained to the observed regions. Different from dynamic NeRFs, we aim to render high-fidelity images of the entire scene at any given timestamp.

The recent progress made in enabling NeRFs to support geometry or appearance editing [23, 14, 15, 47] has been quite significant recently. They focus on editing objects within the scene, for example, changing the color [23, 14], editing the parts of objects [23] or making deformations [15, 44]. This line of work usually handles editing following the user's instruction and is limited to existing objects within the scene while our goal is to adapt NeRFs to various real-world scene changes efficiently.

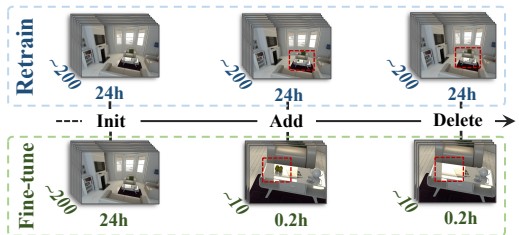

Figure 2: Illustration of the Continual Learning of NeRF.

**Continual Learning** Continual Learning aims to learn different tasks progressively[42, 33]. However, catastrophic forgetting is the key challenge, meaning the model will have performance degradation on previous tasks after training on new task[13]. In our task, the Continual Learning of NeRF is defined as a model that should not have artifacts or quality degradation on unaltered areas after the model has adapted to the scene changes multiple times. (See Figure 2). To tackle the forgetting problem in continual learning, some researchers propose memory replay method[5, 6, 11], which will store the previous training data and use them to augment the current training set while training the model on the new task. Besides, adding regularization constraints to the neural network to limit the parameter updated also proves to be an effective solution[1, 13, 21, 46]. Lastly, Adding separate module to handle different tasks also draw some attentions[18, 32, 34, 45, 25].

## 3 Continual Learning of NeRF

Our study focuses on continual learning for scene-level NeRF, where the goal is to update a pre-trained scene-level NeRF efficiently by utilizing a few new images while retaining the memory of unchanged areas. Figure 2 provides an illustration of the process. In this section, we will first formulate the Continual Learning task of NeRF and subsequently describe our preliminary study which highlights the forgetting issue and motivates our method.

### 3.1 Problem Formulation

We represent a 3D scene as a 5D radiance function, denoted by $F_\theta : (x, d) \rightarrow (c, \sigma)$, in accordance with the vanilla NeRF [26]. Specifically, our approach utilizes an MLP $F_\theta$, where $\theta$ represents a set of parameters, to predict the volume density $\sigma$ and color $c$ according to each sample point $x$ and its corresponding view direction $d$. NeRF has demonstrated a high capacity for high-quality novel view synthesis. Our focus is on the Continual Learning of NeRF to adapt to changes while maintaining the unaltered components.

To simulate natural scene changes, we incorporate basic operations, denoted as $O_p$, including adding, removing, moving, and replacing objects, which are common factors contributing to scene alterations. To emulate these scene changes over time, we simulate a series of operations $[O_{p_1}, O_{p_2}, ..., O_{p_n}]$, representing time-varying changes that result in evolved scenes $[S_{u_1}, S_{u_2}, ..., S_{u_n}]$, respectively. Our simulation aims to capture the following characteristics of real-world scenes: 1) localized changes occurring within objects, 2) the impracticality of extensively recapturing scene-level data in large-scale environments, such as rooms or cities, due to time and cost constraints, and 3) the dynamic evolution of real-world scenes over time.

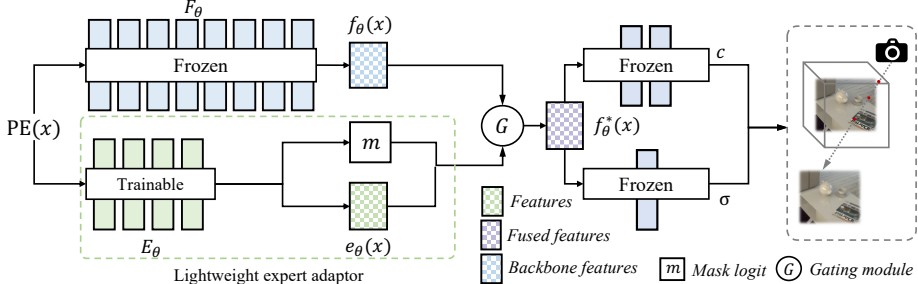

Figure 3: An overall inference framework of CL-NeRF. The position encoding $PE(x)$ is fed into both the original MLP backbone and the proposed lightweight expert adaptor module (the green dashed box), generating backbone features $f_\theta(x)$ and expert features $e_\theta(x)$. A scalar output $m$ by the expert is utilized for feature fusion. Finally, the fused feature $f_\theta^*(x)$ is passed to the volume density and color prediction heads for further volumetric rendering.

Given the changes in scenes over time, our continual learning task is defined as follows: initially, we train a model $F_\theta$ with $m$ multi-view images of an original scene $S_o$. Subsequently, we progressively update $F_\theta$ using $k$ new multi-view images that capture scene changes $S_{u_i}$ after operation $O_{p_i}$ at each time step $i$. It is worth noting that the new training dataset is significantly smaller compared to the original one (i.e., $k \ll m$).

## 3.2 Preliminary Study and Motivations

**Preliminary Study** A straightforward approach to the continual learning task is to directly fine-tune an existing NeRF model with newly acquired data at each instance. We evaluate this baseline using a simple single-operation preliminary experiment. Specifically, as shown in Figure 1 the room-level scene comprises several pieces of furniture. We capture around 200-300 images for the scene to train the original model $F_\theta$. Then, we modify the scene with the ADD operation (i.e., placing several apples on the table) and take 10 images around the object. Finally, we use the new data to fine-tune $F_\theta$ as a baseline.

**Catastrophic Forgetting** However, this simple strategy leads to unsatisfactory performance. While achieving acceptable quality for the updated region after the change, the model results in a catastrophic deterioration of the quality in unaltered regions. As illustrated in Figure 1, after fine-tuning with the new data, artifacts appear on the bookshelf, which is on the unchanged area located on the right-hand side of the table. The average PSNR dramatically drops from **32.71** to **24.21**. We conjecture that NeRF stores geometry and appearance information in the weights of its MLP model in a coupled and implicit manner. Consequently, applying changes locally without affecting other regions is difficult. This issue presents a significant obstacle in adapting pre-trained NeRF models to changing real-world scenes while preserving unaltered areas. The problem is particularly pronounced in large-scale scenes, making recapturing and full retraining impractical in many cases.

**Comparisons with Traditional Continual Learning Task** In practical settings, continually capturing images and retraining a NeRF after every scene change is ideal but unfeasible for large, dynamic scenes. Instead, we propose a more realistic method: fine-tuning existing NeRFs with recent images consistently, which resembles traditional Continual Learning (CL) tasks. This technique, however, can induce artifacts and catastrophic forgetting in previously trained areas, as noted in traditional CL tasks [13]. The divergence between CL in NeRF and traditional CL tasks lies in the inherent conflict between new and old data. While traditional CL tasks maintain a stable one-to-one input-output correspondence, NeRF attributes colors and densities to specific points. Scene alterations can modify these associations, leading to conflicting supervisory signals. Standard CL solutions like Memory Replay (MR) prove insufficient in NeRF due to these inconsistencies, particularly in changed regions. To navigate these challenges, our study introduces a novel method in Section 4.

## 4  Our Method

In this section, we introduce our novel approach, CL-NeRF, for adapting a pre-trained NeRF to scene changes over time. The pipeline comprises two major components: a lightweight expert adaptor specifically designed to adapt to new scene changes and a conflict-aware knowledge distillation

training objective to preserve the memory in unaltered areas (see Section 4.1). As shown in Figure 3, the overall framework involves a pre-trained $F_\theta$ for modeling the entire scene and a small set of new training images $I$ that capture local scene changes. The lightweight trainable expert adaptor, represented as $E_\theta$, generates a feature for capturing scene changes which is further fused with features from the original $F_\theta$ using a masking mechanism. To prevent forgetting, we introduce a conflict-aware self-distillation mechanism (as illustrated in Figure 3) that distills knowledge from $F_\theta$, particularly in unaltered areas. Moreover, we propose a continual learning benchmark for NeRF, which includes various operational definitions and metrics to assess the overall neural rendering performance in adapting to changes and preserving unchanged content (see Section 4.2).

## 4.1 CL-NeRF

**Lightweight Expert Adaptor**   Our approach is motivated by the observation that scene changes primarily occur in localized regions , fine-tuning all the model parameters resulting in forgetting in unaltered regions. To address the above issue, we introduce a lightweight expert adaptor, denoted as $E_\theta$, which can be seamlessly integrated with the original NeRF model, represented by $F_\theta$, as a plug-and-play module. In addition to encoding information about newly modified areas of the scene, $E_\theta$ also provides a rollback strategy that enables users to revert to any desired timestamp. By contrast, $F_\theta$ is responsible for storing information about the overall original scene. These two components offer a comprehensive solution for modeling dynamic scenes with high fidelity.

To achieve our objectives, we define the input and output of $E_\theta$ in the same manner as for $F_\theta$. Specifically, $E_\theta$ takes the same input as the backbone $F_\theta$ and generates two outputs: a mask logit $m$, and a feature vector $f_o$, which is subsequently integrated into the original $F_\theta$. The mask logit $m$ takes on a gating role in the feature fusion module, whereas the feature output by $F_\theta$ is fused with the output of $E_\theta$, as illustrated in Figure 3. This approach enables us to capture dynamic changes in the scene with $E_\theta$ while preserving the overall structure of the original scene with $F_\theta$.

Concretely, each sampling point $x$ is fed to the original network $F_\theta$, which generates a feature $f_\theta(x)$ representing previous contents. Simultaneously, $E_\theta$ predicts a residual feature vector for $x$, represented as $e_\theta(x)$. The final fused feature $f_\theta^*(x)$ can be calculated as

$$f_\theta^*(x) = f_\theta(x) + m \cdot e_\theta(x). \tag{1}$$

Then, $f_\theta^*(x)$ is fed back into the remaining component of $F_\theta$ to further predict volume-density $\sigma$ and view-dependent color $c$ for neural rendering. As $m$ approaches 0, the output features should remain as close as possible to the original features. When $m$ approaches 1, the predicted features from $E_\theta$ will be most effective in predicting modifications made to new areas, also such a design encourages the model not to change the origin scene.

**Conflict-Aware Self-Distillation Mechanism**   The student model, responsible for adapting to a new scenario, is initialized with prior knowledge and fine-tuned to incorporate relevant information from the previous scene. To prevent catastrophic forgetting, we designate the previously trained original model in the preceding scene as the teacher model. Through self-distillation, we transfer scene-related information from the teacher model to the student model. To be specific, we randomly sample camera views and points in space and feed them into both the teacher model and the student model. The teacher model's prediction is used to supervise the student model.

However, the supervision signals provided by the images captured in new camera views can conflict with the supervision from the teacher model. To this end, we propose conflict-aware knowledge distillation. Specifically, for each sampling point, our expert adaptor has already predicted a mask logit. This allows us to utilize volume rendering in the same way as image prediction to render a soft mask map $\hat{M}$. This soft mask map can serve as a probability map indicating the likelihood of a pixel belonging to the changed area. A value close to 1 signifies a high probability of the pixel belonging to the conflict region, in which case the self-distillation mechanism need not be executed. Conversely, when there is a higher probability of the pixel belonging to the unaltered region, calculating an additional knowledge distillation loss to make the supervision signals more accurate. It can be formulated as

$$\mathcal{L}_{\mathrm{kd}} = \|\hat{I}_s \cdot (1 - \hat{M}) - \hat{I}_t \cdot (1 - \hat{M})\|_2, \tag{2}$$

where $\hat{I}_s$ and $\hat{I}_t$ are the images predicted by the student model and the teacher model, respectively.

**Training Objective** Our whole pipeline is trained in a self-supervised manner. The overall loss includes a photometric loss $\mathcal{L}_{\text{ph}}$, a conflict-aware self-distillation loss $\mathcal{L}_{\text{kd}}$ and a mask loss $\mathcal{L}_m$. It is well noted that we introduce a self-distillation training mechanism therefore we can have both the observed new images and a teacher model to provide the supervisory signals. Hence we divide the inputs into two parts based on camera pose distributions, defined as new and old poses, respectively. To be specific, when the input points are sampled using new camera poses, we use $\mathcal{L}_{\text{ph}}$ to generate supervision. $\mathcal{L}_{\text{ph}}$ measures the reconstruction error between the predicted image $\hat{I}_s$ and newly captured ground-truth image $I_n$ as

$$\mathcal{L}_{\text{ph}} = \|\hat{I}_s - I_n\|_2. \tag{3}$$

To further enhance the change awareness, we also propose a mask loss under new poses as

$$\mathcal{L}_{\text{m}} = \|\hat{M} - \hat{M}_{gt}\|_2, \tag{4}$$

where $\hat{M}_{\text{gt}}$ is the pseudo ground-truth mask. $\hat{M}_{\text{gt}}$ is computed by the difference map between newly captured images and the predicted images from the teacher model using the same camera pose as the newly captured images. Besides, when we want to compute $\mathcal{L}_{\text{kd}}$, the uniformly sampled old camera views can be used as inputs to both the teacher model and the student model, and the definition is given in Equation 2. Finally, the overall training objective is

$$\mathcal{L}_{\text{all}} = w_{kd} \cdot \mathcal{L}_{\text{kd}} + w_{ph} \cdot \mathcal{L}_{\text{ph}} + w_m \cdot \mathcal{L}_{\text{m}}. \tag{5}$$

More implementation details are described in the supplementary file.

## 4.2 Benchmarks

To our best knowledge, there is no existing dataset for our proposed continual learning task of NeRF and no proper evaluation metrics to evaluate the forgetting behavior. Therefore, in this section, we will first propose a new dataset containing two synthesis scenes, one real-world scene, and an extra city-level scene. Then we propose metrics specifically designed to evaluate how serious the forgetting after fine-tuning is.

**Operations** We categorize scene operations into three types: atomic, composite, and sequential. *Atomic operations* consist of addition and deletion. *Composite operations* include movement and substitution, which modify positional relationships or replace existing objects with new ones. *Sequential operations* apply a series of basic or composite operations over time. Different from composite operations' focus on quick, specific region modifications, sequential operations aim to monitor long-term error accumulation in changing real-world scenarios.

**Camera Views** We evaluate the reconstruction quality of both local details and the overall scene using two testing indicators: local and global views consisting of a set of evaluation camera poses. Six camera trajectories are designed to cover the entire scene, with five capturing the four orientations of a room (local view) and the sixth simulating a person wandering around the room (global view). Additionally, we define an object-centric camera view to assess adaptation to new changes.

**Task Definitions** We further define the old task as using the camera poses on the six camera trajectories of the global view mentioned above to evaluate the overall quality of the entire scene. While the new task is defined as evaluating the model with an object-centric camera view to represent the local quality of the altered region. In the sequential continual learning setting, we define $n$ tasks ranging from task 1 to task n. Each task corresponds to an operation with an object-centric camera pose centering around the altered region. Which this pose is used in capturing new training data for model adaptation and evaluation. Therefore, to evaluate the forgetting behaviors, we will evaluate all the previous tasks after each fine-tuning. For example, in Table 3, the cell "training on task 4 and testing on task 1" under FT block represents the model after tuning on add, delete, move, and replace data; then evaluate on the add region.

**Evaluation Metrics** PSNR is a well-established rendering quality assessment criterion that measures the peak signal-to-noise ratio between original and reconstructed images. We calculate two quality metrics through the same average process as the local view with a pose from the global view to evaluate the model's memorization of unaltered areas after updating the model. In addition, we have drawn inspiration from the commonly used backward transfer scores employed to assess a continual learning system [24] and have accordingly designed similar metrics for evaluating the forgetfulness

level of our method in the continual learning setting. The forgetting level between the model fine-tuning step $n$ and step $m$, while $n \leq m$, is intuitively measured by the backward transfer model. At each fine-tuning step, a new dataset captured from locally altered regions with corresponding camera poses specific to that tuning step is utilized for model adaptation. It is noteworthy that the pose of each step also serves as the evaluation pose. In such case, we first calculate the $\text{PSNR}_{m,n}$ by using the model tuned at step $m$ to render outputs given poses from step $n$. Then, the definition of the Backward Transfer Metric (BTM) is shown in Eq. (6).

$$\text{BTM} = -\frac{1}{T-1}\sum_{i=1}^{T-1}(\text{PSNR}_{T,i} - \text{PSNR}_{i,i}) \quad \text{and} \quad \text{FM} = \frac{1}{T}\sum_{i=1}^{T}\text{PSNR}_{T,i}. \tag{6}$$

BTM quantifies the level of forgetting by measuring the average cumulative gap between the original model performance ($\text{PSNR}_{i,i}$) and the final performance ($\text{PSNR}_{T,i}$). Furthermore, we introduce a Forgetting Metric (FM) that measures the overall performance upon completion of the entire sequence of operations, with $T$ being the last step. The metric evaluates the average PSNR obtained by having the last model infer all old poses as Eq. (6).

## 5 Experiment

### 5.1 Datasets

We construct two kinds of distinct datasets, synthetic and real, to assess the performance of our algorithm and baselines, specifically focusing on continuous learning and the phenomenon of forgetting, thereby showcasing the algorithm's versatility in diverse environments.

**Synthetic Dataset** Synthetic datasets are created using the Blender simulator, employing predefined meshes with textures. We provide three synthetic datasets: *Dataset Whiteroom*, *Dataset Kitchen* and *Dataset Rome*. The first two datasets encompass indoor, room-level scenes of a living room (depicted in Figure 4) and a kitchen, respectively. Using *Dataset Whiteroom* as an example, for the pre-training stage on the original scene, we capture roughly 200-300 multi-view images from six pre-defined camera trajectories discussed in Section 4.2. The camera poses are designed to encompass the entire scene. To fine-tune the pre-trained model after each operation applied to the original scene with new training data, we collect about 10 multi-view images for each time of change, totaling 40 images for all four operations. Notably, these new images are captured around the modified area's center. Furthermore, we captured all images at 960x640 resolution and partitioned the overall data into an 80% training set and a 20% validation set. The final dataset, *Dataset Rome*, encompasses a large-scale synthetic outdoor environment, featuring a reconstructed mesh of the Colosseum. More descriptions for *Dataset Kitchen* and *Dataset Rome* are provided in the supplementary file.

**Real Dataset** We capture two challenging real-world scenes, *Dataset Lab* (indoor) and *Dataset Court* (outdoor), featuring diverse objects and environments to demonstrate our algorithm's generalizability. For indoor scenes, we use an iPhone 14 Pro to capture both the RGB and LiDAR depth streams via a custom iOS application, with poses determined by ARKit. For outdoor scenes, RGB images are captured with a drone, and poses are computed using COLMAP [37, 38].

We assess the core ATOMIC function ADD. For initial training, we take 100-150 images; for finetuning, an extra 10-20 near changes; to assess adjustments and study task-forgetting, we capture another 10-20 and 20-30 images, respectively, in old and revised areas. We resize all images at a resolution of $960\times720$ and $960\times640$, respectively. More descriptions for *Dataset Lab* and *Dataset Court* are available in the supplementary file.

### 5.2 Baselines for Comparison

As no prior work on our task or related tasks exists, we develop several baselines to evaluate the efficacy of our proposed CL-NeRF. In the supplementary file, we also provide a comparison with the generalizable NeRF.

**Navie Fine-Tuning** Our first baseline is a straightforward fine-tuning approach, denoted as "FT", in which we use new data to fine-tune the NeRF model. In the sequential operation setting, we fine-tune the model at step $t-1$ after applying the $t$-th operation to the scene.

**Memory Replay (MR)** Inspired by the prevalent memory replay approach in lifelong learning [6, 11], we randomly select old training images to augment the fine-tuning training set. Our empirical study

sets the replay data to new data ratio at 50%:50%. In the case of multiple operations appearing in a sequential form, we extend the data memory pool with the new data provided by previous stages. We maintain the same sampling ratio when randomly selecting old data to supplement the training set.

**DyNeRF [17]** We adapt our task to meet the requirements of dynamic NeRF, i.e., DyNeRF[17]. Following its setting, we assign a latent code for data from each operation; this latent code is a trainable vector $z_t$, which is jointly fed into the MLP along with the original input $x$. $z_t$ is a 32-dimensional vector that will be optimized during training. Specifically, in our task, we first pre-train an MLP NeRF with latent code $z_o$ for representing the original scene. At each operation step $t$, we initialize a new latent code $z_t$ and use it to replace the latent code in the trained MLP model at step $t-1$. Then, the whole model is fine-tuned on the new dataset. Furthermore, to prevent forgetting and align the input format of the original DyNeRF, we also employ the same sampling strategy as used in the Memory Replay Baseline. This baseline essentially combines the original DyNeRF [17] and Memory Replay, which sets a strong baseline. Note that the original DyNeRF [17] does not have a mechanism to avoid forgetting.

## 5.3 Main Results

Due to page limitations, we present the results of a single synthetic dataset *Dataset Whiteroom* and qualitative results for several others. Additional results can be found in the supplementary file.

**Single-step Operations** We first compare our method with all three baselines introduced above on single-step operation experiments. As shown in Table 2, our CL-NeRF delivers exceptional performance across all operation types, particularly in ADD and DELETE operations. Our method achieves the best rendering quality on altered regions, shown in the new columns in Table 2 while having the minimum forgetting problem among all the methods. It is important to mention that DyNeRF uses previous-stage old image data in training, while our approach only utilizes new images. We present more results and analysis in the supplementary file.

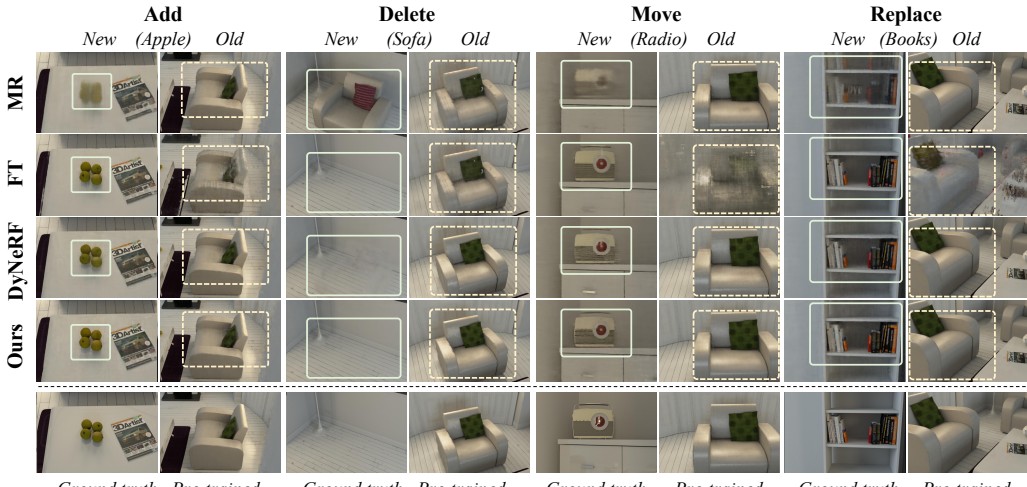

Figure 4: Visualization results of CL-NeRF (Ours) and all compared baselines on 4 single-step operations: add, delete, move, and replace. The ground truth images and inference images from the pre-trained model are used to highlight the rendering quality on new tasks and quality degradation caused by forgetting severity.

Figure 4 demonstrates the algorithm's efficacy in both new and old tasks. The new scene images are rendered from camera poses that reflect scene changes (i.e., object-centric camera pose trajectory), while the old scene images are rendered from camera poses where the scene remains unchanged. Our method captures significantly more details for the newly added object compared to the other three methods. Additionally, our method can completely remove the sofa, while the entire sofa or traces of the sofa remain visible in MR or DyNeRF. In comparison to "FT", our model retains more details of unchanged areas during the delete operation, such as the pillow patterns. In move and replace operations, our model also demonstrates the advantages of preserving unchanged areas, particularly

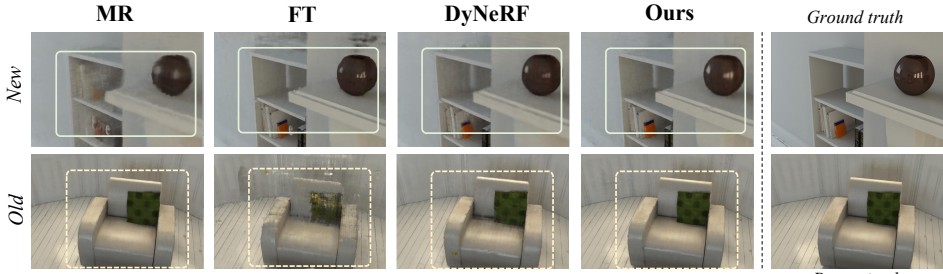

Figure 5: Visualization results after sequential operations. The images in the upper row are rendered after the model has been fine-tuned on four operations, while the images in the lower row test and demonstrate the forgetting issue on the old task.

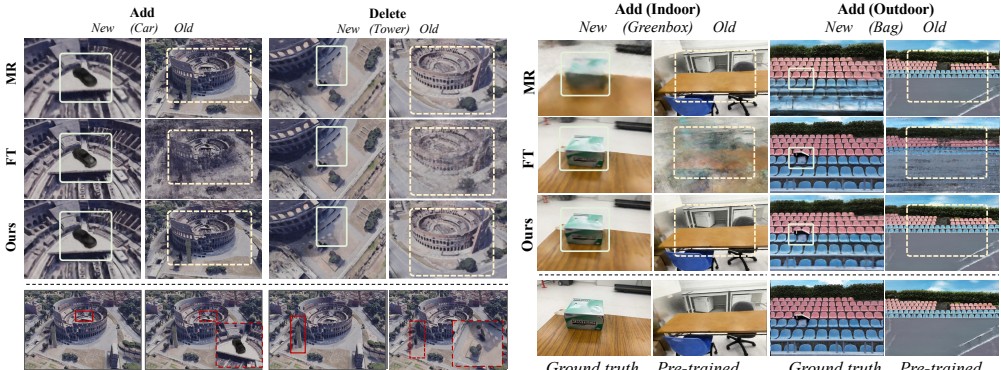

Figure 6: Synthetic *Dataset Rome*.

Figure 7: Real *Dataset Lab/Court*.

in detailed representations. Figures 6 and 7 further attest to our algorithm's adaptability across diverse environments.

| Metrics | FM ↑ | BTM ↓ |
|---------|-------|-------|
| FT | 20.908 | 11.367 |
| MR | 20.930 | **1.233** |
| Ours | **28.638** | 1.707 |

Table 1: Forgetting metrics in sequential operations SEQ on *Dataset Whiteroom*.

**Sequential Operations** Table 1, 2, 3 and Figure 5 present the results of sequential operations. Table 1 employs both BTM and FM to quantify the issue of forgetting, with specifics elucidated in Eq. 6. Table 2 denotes "SEQ" as the sequential operation, evaluating the final model's performance following a sequence of operations, whereas Table 3 examines performance at each individual stage. It can be observed that our proposed method exhibits superior performance in all evaluated tasks compared to the baseline methods. It performs on par with the "FT" baseline in Task 4. This is because both training and evaluation are conducted on Task 4, allowing the fine-tuning baseline to benefit more from the new data. Qualitative comparisons in Figure 5 also demonstrate the effectiveness of our method in delivering high-quality results.

Table 2: Rendering results of atomic/composite/sequential operation on *Dataset Whiteroom*. It showcases the PSNR on both new and old tasks following four single-step operations. The best results are highlighted in **bold**, while the second-best results are underlined.

| Operations | ADD | | DELETE | | MOVE | | REPLACE | | SEQ | |
|------------|------|------|------|------|------|------|------|------|------|------|
| | *Old* | *New* | *Old* | *New* | *Old* | *New* | *Old* | *New* | *Old* | *New* |
| FT | 24.21 | 23.54 | 22.89 | 34.76 | 20.93 | 29.29 | 21.45 | **30.14** | 16.91 | 27.36 |
| MR | 31.97 | 19.40 | 31.40 | 17.21 | 32.39 | 25.66 | 32.48 | 25.15 | 25.95 | 24.45 |
| DyNeRF | 32.29 | 23.89 | 31.39 | 25.82 | 32.83 | **30.12** | 32.70 | 29.88 | 25.80 | **29.83** |
| Ours | **32.33** | **25.29** | **32.43** | **34.77** | **33.30** | 29.97 | **33.33** | 29.82 | **33.38** | 29.74 |

Table 3: Rendering result of the sequential operation on *Dataset Whiteroom*. Tasks 1-4 are defined to evaluate the model based on camera poses corresponding to four operations: add, delete, move, and replace, respectively. Models are evaluated after being tuned for previously completed tasks. The best results are highlighted in **bold**, while the second-best results are underlined.

| Testing on | | FT | | | | MR | | | | Ours | | | |
|---|---|---|---|---|---|---|---|---|---|---|---|---|---|
| | | *Task1* | *Task2* | *Task3* | *Task4* | *Task1* | *Task2* | *Task3* | *Task4* | *Task1* | *Task2* | *Task3* | *Task4* |
| Training on | *Task1* | 23.54 | - | - | - | 19.40 | - | - | - | **25.29** | - | - | - |
| | *Task2* | 20.65 | 34.76 | - | - | 17.48 | 17.21 | - | - | **24.41** | **34.77** | - | - |
| | *Task3* | 16.81 | 21.55 | 29.29 | - | 17.46 | 17.63 | 25.66 | - | **24.05** | **33.78** | **29.97** | - |
| | *Task4* | 14.63 | 20.28 | 18.58 | **30.14** | 17.49 | 17.80 | 23.28 | 25.15 | **23.71** | **33.07** | **28.13** | 29.82 |

## 5.4 Ablation Study

**Expert Adaptor and Distillation** Our model comprises two essential components: the lightweight expert adaptor and the conflict-aware knowledge-distillation learning objective. We conduct an ablation study on these components, with the results shown in Table 4. These results demonstrate that both modules contribute to improving performance. Without the expert adaptor, the model's performance drops significantly when evaluating *new* scenes, highlighting the effectiveness of the expert adaptor in enabling the model to adapt to new changes. In the absence of knowledge distillation (KD), the model performs poorly on *old* scenes, indicating that KD helps address the forgetting problem for the Continual Learning of NeRF.

Table 4: Ablation Study on *Dataset Whiteroom*. We study the importance of the expert adaptor and distillation strategy.

| Operations | ADD | | DELETE | | MOVE | | REPLACE | | SEQ | |
|---|---|---|---|---|---|---|---|---|---|---|
| | *Old* | *New* | *Old* | *New* | *Old* | *New* | *Old* | *New* | *Old* | *New* |
| **w/o Expert** | 31.24 | 23.42 | 30.36 | 27.48 | 31.23 | 30.86 | 31.33 | 30.00 | 29.39 | 24.30 |
| **w/o KD** | 29.81 | 24.10 | 29.27 | 34.57 | 23.90 | 29.92 | 23.47 | 28.97 | 18.63 | 28.19 |
| **w/o $L_m$** | 31.08 | 24.92 | 30.99 | 31.73 | 31.90 | 28.64 | 32.63 | 28.53 | 30.70 | 28.05 |
| **Ours** | **32.33** | **25.29** | **32.43** | **34.77** | **33.30** | **29.97** | **33.33** | **29.82** | **33.38** | **29.74** |

**Mask Loss.** In order to demonstrate the necessity of mask loss in resolving conflict, we conduct an experiment with mask loss $L_m$ removed. As shown in Table 4, our findings suggest that mask loss $L_m$ also contributes to high-quality view syntheses. Without the guidance provided by the mask loss, the model may struggle to correctly predict object boundaries and often produce inaccurate results.

## 6 Conclusion

We have presented a method CL-NeRF for efficiently adapting pre-trained NeRF to changing environments. CL-NeRF comprises two major components: a lightweight expert adaptor for modeling new changes in evolved scenes, and a conflict-aware knowledge distillation strategy for preserving memory in unaltered areas. These two designs collaborate, enabling our method to achieve superior results. To facilitate future research, we have also introduced a continual learning benchmark for NeRF with comprehensive evaluation metrics. To the best of our knowledge, this is the first of its kind. We hope our investigation and comprehensive benchmark inspire further research in this task, empowering NeRF to effectively model the ever-changing real-world scenes. More discussions about broader impacts, limitations including the scalability and other parameter-efficient tuning (PET) method are described in the supplementary file.

### Acknowledgments

This work has been supported by Hong Kong Research Grant Council - Early Career Scheme (Grant No. 27209621), General Research Fund Scheme (Grant No. 17202422), and RGC Matching Fund Scheme (RMGS), and Tencent Research Fund.

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
