# OpenReview forum: "CL-NeRF: Continual Learning of Neural Radiance Fields for Evolving Scene Representation"
_NeurIPS.cc/2023/Conference — NeurIPS 2023 poster_

### Official Review · Reviewer_Fafa · 2023-06-29

**Soundness:** 3 good
**Presentation:** 3 good
**Contribution:** 3 good
**Rating:** 5
**Confidence:** 4

**Summary:**

The paper introduces CL-NeRF, an approach for efficiently adapting Neural Radiance Fields (NeRFs) to real-world scene changes over time. CL-NeRF focuses on continual learning and requires only a few new images to adapt to changes while retaining the memory of unaltered areas. It consists of two main components: a lightweight expert adaptor for adapting to new changes and evolving scene representations, and a conflict-aware knowledge distillation learning objective for memorizing unchanged parts. The authors also propose a new benchmark with comprehensive metrics to evaluate Continual Learning of NeRFs.

**Strengths:**

1. The inclusion of a benchmark is a valuable contribution as it provides a standardized evaluation framework for assessing the performance of continual learning methods applied to NeRFs.
2. The provision of rendered data sets generated by CL-NeRF is beneficial for the research community.

**Weaknesses:**

The paper's weakness lies in the assumption that continual learning is necessary for NeRFs and the relevance of this assumption to the problem at hand. NeRFs have distinct characteristics that differentiate them from traditional machine learning problems. Firstly, NeRFs do not typically require a vast amount of training data, as they focus on scene-specific recon rather than generalization (i.e., the plain NeRFs studied in the paper, not SRN or pixelNeRF). One scene -- one NeRF.
Additionally, NeRFs do not necessarily face the issue of inaccessible training data, as the training data can be obtained by rendering images of the scene. Therefore, the need for continual learning, which assumes the unavailability of training data, may not be applicable to NeRFs.

Furthermore, the paper does not adequately address a simple baseline approach __for the scene change challenge__. By rendering images for the new given poses, it becomes straightforward to identify areas where the scene changes occur. This straightforward method eliminates the need for complex continual learning techniques in this specific scenario.


---

After rebuttal:

I have updated my score to be borderline accept. The authors' response regarding large-scale tuning of a trained NeRF model has convinced me of the value of their proposed method. While the experiments do not directly validate performance in the large-scale scenario, the current results demonstrate promising capabilities that could benefit future research in this direction. All other reviewers value the contribution of the continual learning nerf method in this paper.

However, I still have some comments about the writing of the paper to avoid potentially misleading readers on the concept of continual learning:

1. The mentioning of "catastrophic forgetting" may require more explanation. Most readers likely associate this problem with the paper "Overcoming catastrophic forgetting in neural networks," [1] which highlighted it as a serious challenge for sequential learning tasks. But NeRF does not assume sequential learning, since all previous data is stored. The authors could emphasize that even with full data retention, fine-tuning can still be expensive without a continual learning scheme.

2. It may be helpful to include some discussion of efficient network tuning methods as motivation, like the well-known LoRA scheme. Although this paper uses MLPs, parameter-efficient tuning techniques for other networks like transformers are highly relevant to the problem setting.

[1] Kirkpatrick, James, et al. "Overcoming catastrophic forgetting in neural networks." Proceedings of the national academy of sciences 114.13 (2017): 3521-3526.

**Questions:**

1. Why do we need to assume that the training data is not available? The training images themselves are already memorized by NeRFs.
2. Why not just compare the image before change and after change?

**Limitations:**

The problem setting is problematic.

---

> ### Author Rebuttal · Authors · 2023-08-09
>
> Dear Reviewer Fafa,
>
> Thank you for appreciating our approach. We will address your comments below.
>
> **W1: The assumption that continual learning is necessary for NeRFs and the relevance of this assumption to the problem.**
>
> Thank you for the comment. We understand your concerns regarding continual learning for NeRFs. However, we believe that there are potential scenarios where continual learning is urgently needed:
>
> 1) For large-scale scene applications, like city-level scene reconstruction/novel view synthesis, capturing images can be cumbersome and require expensive devices such as drones to capture the entire city. Thus, a setting that requires fewer images for training would alleviate the burden of data capture.
>
> 2) As changes may occur frequently, it would be time-consuming to gather training data and retrain the model after each change. For example, if there are ten scene changes, capturing the entire scene ten times is prohibitively expensive. The burden is further heightened by the need for camera calibration and pose estimation, especially when dealing with a large number of images.
>
> 3) Our investigation may also be useful for resource-constrained platforms, such as smartphones, where storage resources are an important consideration. This issue is bypassed by our continual learning formulation, as we only need to retain the few images that reflect scene changes.
>
> Moreover, the primary challenge that our study addresses lies in adapting to changes without encountering conflicting supervisory signals, rather than whether we can use information from a pre-trained NeRF or the amount of training data available. Even when the data are available, our designs, especially the adaptor and conflict-aware mechanism, are still useful.
>
> We specifically address the challenge of learning new content with conflict in our paper. By using our method, we can adapt to changes and maintain the integrity of scene representations, enabling us to adjust to scene changes with minimal effort (i.e., 10-20 training images and 10-20 minute training time).
>
> **W2: Does not adequately address a simple baseline.**
>
> In fact, we have employed the straightforward method you mentioned, as detailed in Section 4.1, to identify and generate the pseudo mask for supervising the training of the mask logit. However, the main objective of our continual learning task is to learn a new neural radiance field from the new data that captures scene changes.
>
> Simply identifying areas where scene changes occur does not address the problem of adapting the neural radiance field to the new scene content while maintaining unaltered scene representation. Our proposed continual learning techniques aim to provide a more effective solution to this challenge by leveraging the newly captured images and minimizing the forgetting of previously learned content.
>
> **Q1: Why do we need to assume that the training data is not available?**
>
> Please refer to the question **W1: The assumption that continual learning is necessary for NeRFs and the relevance of this assumption to the problem**.
>
> **Q2: Why not just compare the image before change and after change?**
>
> Please refer to the question **W2: Does not adequately address a simple baseline**.

---

> > ### Comment · Reviewer_Fafa · 2023-08-10
> >
> > Thanks for your detailed explanation.
> >
> > > For large-scale scene applications, like city-level scene reconstruction/novel view synthesis, capturing images can be cumbersome and require expensive devices such as drones to capture the entire city.
> >
> > This makes sense to me. (But I recommend the authors to add this kind of illustrations in the paper to highlight this insight. Currently, the indoor scene is a little misleading.)
> >
> > This concern again aligns with the criticism raised by Reviewer f2t1 regarding the validation: __The number of scenes in the experiment is not large and all of them are synthetic.__
> >
> > The claim is handling large scale scenes with a lot of images, but all results are on a toy dataset. But I agree that validating the ideas could still be of value, I will see how other reviewers comment on this and join the majority.
> >
> > > Moreover, the primary challenge that our study addresses lies in adapting to changes without encountering conflicting supervisory signals, rather than whether we can use information from a pre-trained NeRF or the amount of training data available.
> >
> > I cannot really get the meaning of this challenge. What does `without encountering conflicting supervisory signals` mean here? Do you mind explaining a little bit further? Thanks.
> >
> > > In fact, we have employed the straightforward method you mentioned, as detailed in Section 4.1, to identify and generate the pseudo mask for supervising the training of the mask logit.
> >
> > Apologize if I didn't make it clear. I mean that for experiments in Tab 2, Fig 4,5, the baseline can be discussed.
> >
> > > Simply identifying areas where scene changes occur does not address the problem of adapting the neural radiance field to the new scene content while maintaining unaltered scene representation.
> >
> > We can just sample points from those unchanged areas, and add a loss to ask the MLP outputs unchanged.

---

> > > ### Author Response · Authors · 2023-08-12
> > >
> > > Reviewer Fafa,
> > >
> > > Thanks for your response and suggestions!
> > >
> > > > This makes sense to me. (But I recommend the authors to add this kind of illustrations in the paper to highlight this insight. Currently, the indoor scene is a little misleading.) This concern again aligns with the criticism raised by Reviewer f2t1 regarding the validation: The number of scenes in the experiment is not large and all of them are synthetic. The claim is handling large scale scenes with a lot of images, but all results are on a toy dataset. But I agree that validating the ideas could still be of value, I will see how other reviewers comment on this and join the majority.
> > >
> > > 1) We conduct experiments on two more challenging real-world indoor and outdoor scenes containing various objects and environments to demonstrate the generalization ability of our algorithm. For the indoor scene, we capture images using a camera with LiDAR (specifically, an iPhone 14 Pro) and employ the Record3D App with ARKit to ensure accurate camera poses. On the other hand, the outdoor scene is captured using a DJI drone, focusing on a large-scale subject (the audience seats on a basketball court), with COLMAP utilized to calculate the camera poses. The results (Table R1, Figure R1 in the provided PDF file) reveal that Fine-tuning suffers in old tasks, and Memory Replay struggles with new tasks, highlighting our method's robustness and adaptability across varying environments.
> > >
> > > 2) Owing to time limitations within the rebuttal period, our research has not been extended to encompass city-scale real-world scenes at this time. Despite this constraint, we remain confident that our methodologies and findings hold potential for practical applications. In the future, we intend to collaborate with industry partners to explore expanding our research to broader and more comprehensive scales.

---

> > > ### Author Response · Authors · 2023-08-12
> > >
> > > > I cannot really get the meaning of this challenge. What does without encountering conflicting supervisory signals mean here? Do you mind explaining a little bit further? Thanks.
> > >
> > > In the task of adapting a pre-trained NeRF to the scene changes, conflict is inherent. Utilizing both old training data and newly acquired or estimated images from a pre-trained NeRF, conflicts can emerge within the altered region(e.g., a newly added apple). We have tested in these two cases:
> > >
> > > 1) **Case 1 - Utilizing Old Training Data**: Our baseline algorithm, MR, trains on both original and new data without handling conflicts. Results are shown in Tables 1-3 and Figures 4-5 of the main paper, with further details in the supplementary materials and the provided PDF.
> > >
> > > 2) **Case 2 - Leveraging Pre-trained NeRF**: Our ablation study **w/o Expert** trains on new data and uses the KD strategy with a pre-trained NeRF without addressing conflicts. Results are presented in Table 4 in the main paper.
> > >
> > > From the results, we can conclude that without properly handling conflicting signals, new task performance in altered regions is unsatisfactory. To address the conflict problem, we introduce a conflict-aware knowledge distillation method, effectively utilizing information from both old training data and a pre-trained NeRF, resulting in a remarkable performance.

---

> > > ### Author Response · Authors · 2023-08-12
> > >
> > > > Apologize if I didn't make it clear. I mean that for experiments in Tab 2, Fig 4,5, the baseline can be discussed.
> > >
> > > It appears there may be some misunderstandings. Our lightweight expert adaptor can identify areas of scene changes, aligning with the method you referenced. Hence, the results are already addressed in Table 2, Figure 4, and Figure 5.
> > >
> > > Specifically, our expert predicts a mask logit for each sampling point to indicate if it has been altered (with a logit close to 1 for a newly added region). Using this logit, we fuse original and new predicted features, as outlined in Equation 6 and Figure 3, alongside a mask loss defined in Lines 212-214. This mask loss can further assist the training process for mask logit estimation.

---

> > > ### Author Response · Authors · 2023-08-12
> > >
> > > > We can just sample points from those unchanged areas, and add a loss to ask the MLP outputs unchanged.
> > >
> > > Sampling points from unchanged areas can be divided into two cases:
> > >
> > > 1) **Case 1**: If we attempt to sample points from unchanged areas in old images, we must first define which regions within the old images have changed. This identification may not be feasible. Directly using old training data will result in conflict in the altered region, as we discussed above.
> > >
> > > 2) **Case 2**: If we rely on sampling points from unchanged areas in new images, a different problem arises. Given that the number of new images is typically limited, the model may face difficulties in effectively mitigating the forgetting phenomenon using this restricted data.
> > >
> > > We further explore Case 2 with an experiment using pseudo mask ground truth, denoted $\hat M_\text{gt}$. Our fine-tuning process for the pre-trained NeRF includes calculating two losses. First, we sample points from the changed areas, calculating the photometric error between the predicted image and the newly captured images. Second, we sample points from the regions that remain unchanged, and we expect the MLP outputs to remain consistent. The experiment is carried out on the Whitehouse dataset, utilizing the ADD operation. The results below compare our algorithm with and without the proposed lightweight expert adaptor (i.e., the same structure as the pre-trained NeRF). The findings indicate that this simple masked sampling strategy results in unsatisfactory performance in the old task.
> > >
> > > | Algos| Old| New|
> > > |--------|--------|--------|
> > > |Masked Sampling | 28.97|23.64 |
> > > |Ours w/o Expert| 31.24 | 23.42 |
> > > |Ours | **32.33** | **25.29** |

---

> > > > ### Comment · Reviewer_Fafa · 2023-08-17
> > > >
> > > > Thank you for the detailed reply. That makes sense to me and I have read other reviewers comments -- happy to see that all of others value your contribution in continual learning nerf.
> > > >
> > > > I updated my comments in the weakness comments: Please consider this additional feedback, as it may help clarify your contributions.

---

> > > > > ### Author Response · Authors · 2023-08-20
> > > > >
> > > > > Dear Reviewer Fafa,
> > > > >
> > > > > Thank you for your thoughtful feedback and questions. We're grateful for your acknowledgment. Our detailed responses are provided below.
> > > > >
> > > > > **W1 (after rebuttal): The mentioning of "catastrophic forgetting" may require more explanation.**
> > > > >
> > > > > We would like to add more explanations about the concept and our motivations.
> > > > >
> > > > > In our opinion, we view fine-tuning NeRF multiple steps as a sequential learning task. Thus, we are the first to integrate Continual Learning (CL) with NeRF. Ideally, in real-world applications, one could capture new images and retrain a NeRF after every scene change. However, due to the impracticality of this approach in large, dynamic scenes, a more realistic strategy is to fine-tune an existing NeRF using a few recent images. This adaptation resembles traditional CL tasks, where each scene alteration is a sequential task. Our research indicates that this method can introduce significant artifacts in previously trained regions, like catastrophic forgetting in the traditional CL task [1].
> > > > >
> > > > > However, CL in NeRF indeed differs from traditional CL tasks due to inherent conflicts between old and new data. In standard CL tasks like classification or generation, stable one-to-one mapping between input and output exists. Conversely, NeRF assigns color and density values to specific query points. When a scene undergoes changes, these attributes associated with a query point can shift, creating conflicting supervisory signals. Whereas techniques like Memory Replay (MR) suffice in classic CL, they can produce contradictory feedback in NeRF, especially within altered areas. As a result, our study offers specialized approaches such as the lightweight expert adaptor and the conflict-aware KD mechanism to address these challenges.
> > > > >
> > > > > Thank you for your constructive feedback. We sincerely appreciate your insights and will integrate these clarifications into the final version of our paper.
> > > > >
> > > > > **W2 (after rebuttal): It may be helpful to include some discussion of efficient network tuning methods.**
> > > > >
> > > > > We have previously experimented with LoRA on the  Dataset Whitehouse with the ADD operation and found it yielded suboptimal results (especially details preservation in visualization), as detailed in the table below. One plausible explanation is that LoRA may not have sufficient capacity to capture scene changes effectively. In our implementation, we replace the expert by integrating a LoRA module with every fully-connected layer in the pre-trained NeRF, setting its low intrinsic dimension to 16. LoRA's design is to mimic the low intrinsic dimension by setting a bottleneck dimension $r$ which is usually smaller than the original layer's dimension. It achieves this using a downward projection matrix $A$ sized $[d,r]$ and an upward projection matrix $B$ sized $[r,d]$, with $d$ being the original layer's dimension (256 in our case). Notably, there's an absence of nonlinear activations between these matrices. While these designs reduce the parameter count, they may also compromise the model's adaptability. As a result, our current expert utilizes a compact MLP consisting solely of fully-connected layers. Nevertheless, integrating parameter-efficient tuning (PET) into NeRF, with a refined design approach, presents an intriguing future research avenue given its efficiency in fine-tuning fewer parameters. Thank you for your valuable feedback. We truly appreciate your insights and will incorporate them into our paper's final version.
> > > > >
> > > > > | Algorithms| Old | New |
> > > > > |--------|--------|--------|
> > > > > | LoRA | 31.69 | 24.86 |
> > > > > | **Ours (Expert)** | **32.33** | **25.29** |

---

### Official Review · Reviewer_tKPz · 2023-07-05

**Soundness:** 3 good
**Presentation:** 3 good
**Contribution:** 2 fair
**Rating:** 5
**Confidence:** 4

**Summary:**

This paper tackles the task of continuing learning of NeRF, which aims to adapting NeRFs to real-world scene changes over time using a few new images. To prevent the forgetting problem during adapting, the authors propose CL-NeRF. The CL-NeRF consists of two key components: an expert adaptor for adapting to scene changes and a conflict-aware distillation scheme for memorising unchanged parts. The expert adaptor learns to encode local changes  and can be learned from just a few new images. The conflict-aware distillation scheme is designed to preserve the original scene representation for unchanged areas by via student-teacher knowledge distillation. Moreover, A new continual learning benchmark for NeRF is introduced to evaluating the proposed method.

**Strengths:**

1. This paper tackles a practical task, which aims to adapt NeRFs to real-world scene changes over time with minimal data.
2. The paper is well written and easy to follow.
3. New datasets and evaluation metrics are proposed for the introduced task.


**Weaknesses:**

1.	The experimental results are not very well represented. For example, there is no text explanation for Table 1.  What does the ‘SEQ’ represent in Table 2 since there are also results for sequential operation in Table 3. The quantitative results are shown without clear text illustration.
2. Follow the previous comment, the qualitative results in Figure 4 are not very intuitive.  For example, the new and old scenes are very different after the adding operation in the first column. From my understanding, it should be only adding the apple to the old scene?  Similar for the Move and Replace operation, the new and old scenes seem very different.
3.	The predicted mask plays an important role in detecting changing area, it would be better to show some qualitative or quantitative results on how well the mask logits are predicted.
4. The scale of the network will increase when more operation are added since the expert adaptor is scene-specific.


**Questions:**

1.	What is the performance gap between the proposed method and the scenario when new data is abundant? The latter scenario is upper bound for the proposed task.
2.	How many testing data are used for each operation? Take the ‘ADD’ operation for example, we can add different objects to different scenes. Is the result based on just one or multiple scenarios?

**Limitations:**

The limitations and potential negative societal impact are resolved in the supplementary material.

---

> ### Author Rebuttal · Authors · 2023-08-09
>
> Dear Reviewer tKPz,
>
> Thank you for appreciating our approach. We will address your comments below.
>
> **W1: Experimental results are not very well represented.**
>
> 1) We regret the omission of the analysis for Table 1 and appreciate your comments. Table 1 uses the Backward Transfer Metric (BTM) and Forgetting Metric (FM), which are both employed to quantify the forgetting issue. BTM quantifies the level of forgetting by measuring the cumulative gap between the original model performance ($PSNR_{i,i}$) and the final performance ($PSNR_{T,i}$), whereas FM directly measures actual performance after T operations. Details are explained in Section 4.2 (evaluation metrics) in our paper.
>
> 2) SEQ stands for sequential operation, as defined in Section 4.2. In both Tables 2 and 3, 'SEQ' refers to the same operation. However, we assess this operation from two different perspectives. In Table 2, we evaluate the performance (i.e., PSNR) of the final model after a series of operations, while Table 3 assesses the performance at each stage. This distinction provides a more comprehensive understanding of our method's performance throughout the entire process.
>
> **W2: Qualitative results in Figure 4 are not very intuitive.**
>
> Yes, only adding an apple to the old scene. The difference in qualitative results comes from the different camera views. The new scene images are rendered from camera poses that reflect scene changes (i.e., object-centric camera pose trajectory), while the old scene images are rendered from camera poses where the scene remains unchanged. They belong to different areas; thus, we render them to depict the effective adaptation of new scenes and the preservation of unchanged areas. We sincerely apologize for any confusion and will clarify this point in the final version of our paper.
>
> **W3: How well the mask logits are predicted?**
>
> We have included a visualization of the predicted mask in Figure R4 to demonstrate its quality. A pixel value approaching 1 (white) indicates a high probability that the pixel belongs to an altered region, whereas a lower value suggests an unaltered region. This visualization offers a qualitative insight into the effectiveness of our mask logits in detecting changes within the scene.
>
> **W4: The scale of the network will increase when more operation are added.**
>
> We would like to clarify that we employ only one expert throughout our process. For task-1, no experts are used. At task-2, one expert is added. For task-3 and subsequent tasks, we continue with just one expert, utilizing the original model combined with the expert from the previous task to distill knowledge into the current expert. This approach allows our method to efficiently adapt to new tasks while preserving knowledge from previous tasks without the need to add multiple experts.
>
> In our current exploration, this strategy is sufficient to achieve reasonable performance (see Table 1-3 in the main paper) and outperform baseline methods without increasing model complexities. We believe that adding one expert at a time may further boost performance, albeit at the cost of increased model complexity. We will study the trade-off between model complexities (adaptor numbers) and performance in the final version of our paper.
>
> **Q1: What is the performance gap when new data is abundant?**
>
> To demonstrate the upper bound of our proposed task, Figure R5 shows the relationship between performance and the number of images, specifically for the ADD operation on the Whitehouse dataset. The figure reveals that while increasing the number of training images does enhance performance, the improvement is relatively minor.
>
> **Q2: How many testing data for each operation? Add different objects to different scenes? Is the result based on just one or multiple scenarios?**
>
> 1) For synthetic datasets such as Whitehouse, Kitchen, and Rome, the testing data comprises approximately 20 images.
> 2) In our study, we primarily focus on basic operations involving a single object. Additionally, we have conducted tests with multiple objects added simultaneously, and the results are presented in Table R3 and Figure R3.
> 3) Results in each table correspond to each specific scenario, however we encompass the tests across several synthetic and real-world datasets.

---

> > ### Comment · Reviewer_tKPz · 2023-08-18
> >
> > Thanks for the authors' responses. Most of my concerns are resolved in the rebuttal and I would keep my original rating as borderline accept.

---

### Official Review · Reviewer_f2t1 · 2023-07-06

**Soundness:** 3 good
**Presentation:** 3 good
**Contribution:** 3 good
**Rating:** 6
**Confidence:** 5

**Summary:**

This paper proposes a challenge of how to reduce the data and time cost of retraining NeRF when the scene changes. To this end, the paper proposes two key components: a trainable network for the changing part of the scene, and a conflict-aware network for the unchanged part of the scene. With these two components, CL-NeRF achieves high training efficiency for updating the changing part of the scene while preserving the unchanged part well. This leads to the proposal of a new benchmark based on this challenge.

**Strengths:**

* When there are changes in the scene, such as adding or removing objects, CL-NeRF can update the previously modeled results with very few photos and remember the unchanged parts.
* By using a lightweight expert adapter to predict masks, CL-NeRF can perceive which parts have been changed and only modify the features related to the location of the scene change. The self-distillation mechanism that is aware of conflicts can reduce conflicts in the supervised signals.
* A new dataset has been established to evaluate under the condition of scene changes.

**Weaknesses:**

* The number of scenes in the experiment is not large and all of them are synthetic. However, this is understandable due to limitations in time and equipment. It is highly recommended to evaluate the method on more real scenes.
* Training DyNeRF using Memory Replay (MR) methods and simultaneously using photos with both changed and unchanged scenes may lead to ambiguity, which may lead to unfair comparisons. A better MR method may be partitioning the data based on whether the region has changed or not.

**Questions:**

* It is possible that the poor performance of DyNeRF in scene representation is due to the use of MR methods. If this is the case, how will the performance of DyNeRF be when using only the images of the unchanged parts of the scene and the images of the changing parts separately?
* The experiment only includes a few scenes, all of which are synthetic. There are no experiments with real scenes. It would be beneficial to include experiments with more scenes in future studies.

**Limitations:**

The authors have addressed limitations in the supplementary material.

---

> ### Author Rebuttal · Authors · 2023-08-09
>
> Dear Reviewer f2t1,
>
> Thank you for appreciating our approach. We will address your comments below.
>
> **W1: Evaluate the method on more real scenes.**
>
> We conduct experiments on two more challenging scenes (real-world indoor and outdoor scenes) containing various objects and environments to demonstrate the generalization ability of our algorithm.
>
> For the indoor scene, we capture images using a camera with LiDAR (specifically, an iPhone 14 Pro) and employ the Record3D App with ARKit to ensure accurate camera poses. On the other hand, the outdoor scene is captured using a DJI drone, focusing on a large-scale subject (the audience seats on a basketball court), with COLMAP utilized to calculate the camera poses. The results of these experiments are presented in Table R1 and Figure R1. We observe that Fine-tuning (FT) exhibits severe forgetting in old tasks, while Memory Replay (MR) underperforms in new tasks, particularly in qualitative results. These findings effectively showcase the robust performance of our methods across varying environments, further demonstrating the versatility and adaptability of our proposed algorithm.
>
> **W2: Training DyNeRF using a better MR method.**
>
> 1) It is important to note that the original DyNeRF does not include a memory replay (MR) operation as our method does. DyNeRF performs poorly without any MR, as discussed in section B.4 of the supplementary materials. Therefore, in Table 2 of the main paper, we compare DyNeRF with a robust MR method that retains all previous-stage data.
>
> 2) Following your suggestion, we also explore more accurate MR methods on DyNeRF by manually excluding the conflict views, with results presented in Table R4. MR1 represents the original MR method from our main paper, while MR2 excludes frames with conflict regions in the old task, which have been manually removed. This manual selection effectively avoids conflict, enhancing new task performance but reducing the original image count, thus weakening old task performance.
>
> 3) Our method demonstrates superior performance compared to DyNeRF with various MR strategies. Importantly, our approach does not rely on using any previously captured images for training, whereas DyNeRF with the augmented MR method requires storing old images from all previous stages. This distinction highlights the efficiency and effectiveness of our proposed method in handling changing scenes without the need for extensive storage and manual intervention.
>
> **Q1: How will the performance of DyNeRF be when using data without conflict?**
>
> Please refer to the question **W2: training DyNeRF using a better MR method**.
>
> **Q2: It would be beneficial to include experiments with more scenes.**
>
> Please refer to the question **W1: evaluate the method on more real scenes**.

---

> > ### Comment · Reviewer_f2t1 · 2023-08-21
> >
> > Thanks for the authors' response, which has resolved most of my concerns. It is highly recommended to supplement the additional contents to the final version. Besides, dynamic NeRF methods are one of the most important streams of related works, but some recent representative dynamic NeRF works are missing, including but not limited to HyperNeRF, Hexplane, K-planes, TiNeuVox, dycheck *etc*. It is suggested to have a careful check and include the missing works.

---

> > > ### Author Response · Authors · 2023-08-21
> > >
> > > Dear Reviewer f2t1,
> > >
> > > thank you for recognizing our approach. We sincerely apologize for overlooking the related work and acknowledge the importance of dynamic NeRF in our study. We will address this in our final version.
> > >
> > > In our original version, we categorize dynamic NeRF [1-5,8] works into two groups: scene flow (or deformation) estimation, and time-aware design. However, given that some time-aware approaches also utilize time for deformation estimation, we now categorize these researches into two streams: deformation field estimation and the incorporation of time-aware inputs.
> > >
> > > HyperNeRF [6] and TiNeuVox [7] are categorized under deformation estimation since they estimate deformations to a canonical space. Hexplane [9] and K-planes [10], which introduce time-aware features from specific planes, fall under the time-aware inputs incorporation category. Additionally, Dycheck [11] offers a reality check on these dynamic NeRF works.
> > >
> > > Here are the revisions and will incorporate them into our final version:
> > >
> > > This line of work usually takes videos containing dynamic objects as inputs and can be divided into two lines: deformation field estimation [1-7] and the incorporation of time-aware inputs [8-10]. Dycheck [11] offers a critical assessment of recent developments in dynamic NeRFs.
> > >
> > > [1] Neural scene flow fields for space-time view synthesis of dynamic scenes.
> > >
> > > [2] Dynibar: Neural dynamic image-based rendering.
> > >
> > > [3] Nerfies: Deformable neural radiance fields.
> > >
> > > [4] D-nerf: Neural radiance fields for dynamic scenes.
> > >
> > > [5] Nerfplayer: A streamable dynamic scene representation with decomposed neural radiance fields.
> > >
> > > [6] HyperNeRF: A Higher-Dimensional Representation for Topologically Varying Neural Radiance Fields.
> > >
> > > [7] TiNeuVox: Time-Aware Neural Voxels.
> > >
> > > [8] Neural 3d video synthesis from multi-view video.
> > >
> > > [9] HexPlane: A Fast Representation for Dynamic Scenes.
> > >
> > > [10] K-Planes: Explicit Radiance Fields in Space, Time, and Appearance.
> > >
> > > [11] Monocular Dynamic View Synthesis: A Reality Check.

---

> > > > ### Comment · Reviewer_f2t1 · 2023-08-22
> > > >
> > > > Thanks for your response. It would be good to incorporate this into the final version.

---

### Official Review · Reviewer_fqk8 · 2023-07-08

**Soundness:** 4 excellent
**Presentation:** 3 good
**Contribution:** 3 good
**Rating:** 7
**Confidence:** 4

**Summary:**

The authors propose CL-NeRF, which tries to solve the problem of rendering scenes that evolve over time using a few images of the altered scene while retaining information about the unaltered regions. The proposed method contains two key components - 1. an expert adapter, to adapt to new regions, and 2. a conflict-aware knowledge distillation to preserve the original scene representation in the unchanged regions. The paper also introduces a new benchmark to evaluate NeRF-based methods on a continually evolving scene, particularly focusing on a few key operations including ADD, REMOVE, MOVE, and DELETE. The method claims strong adaptation to scene changes, requiring minimal images captured in the changed area while still ensuring high rendering quality in the unchanged regions.

**Strengths:**

- The proposed task is novel, i.e. given sufficient images at time t=0 to train a NeRF, we want to adapt the radiance field (at time t=t > 0) to new changes in the scene with as few images in the altered regions.
- The paper is easy to read and follow. All the components in the proposed method are well-motivated.
- The authors introduce a new continual learning benchmark for NeRFs, containing 3 scenes with 4 single-step operations and a sequence of them. The method is compared against sufficient baselines, outperforming them by a sufficient margin.

**Weaknesses:**

- There are not many significant weaknesses, however, some components of the method e.g. knowledge distillation to prevent forgetting have been seen in some form (without being conflict aware) in [1]. This does not decrease the novelty of the current work, but it would be interesting to evaluate these methods in the proposed benchmark.
- It would also be interesting to see how this method would compare against a Generalizable NeRF baseline that uses the memory replay of the previous task (perhaps a bit more selective to not cover the altered regions) and the new training images captured in the altered region.

1. MEIL-NeRF - Memory Efficient Increment Learning of Neural Radiance Fields.

**Questions:**

- In the sequential setup, are new experts added to the network from the previous task? i.e at task-1 there are no experts, task-2: 1 expert is added, task-3: 2 experts are added, etc.
- Is it necessary for the captured images in the later tasks to be local? i.e. only capturing the altered regions? What would happen if the captured images are more global and covered the entire scene?
- In the current benchmark, every operation only manages to alter one area. It would be interesting to evaluate multiple altered regions and how the current method would scale in such cases.

**Limitations:**

I don't see any negative social impact on their work. The authors have discussed limitations in the supplementary.

---

> ### Author Rebuttal · Authors · 2023-08-09
>
> Dear Reviewer fqk8,
>
> Thank you for appreciating our approach. We will address your comments below.
>
> **W1: Evaluate [1] in the proposed benchmark.**
>
> Thank you for the insightful advice. Unfortunately, due to the limited rebuttal time and the unavailability of MEIL-NeRF's [1] code, it is challenging for us to reproduce their work and provide a comparison in this response. Nevertheless, we appreciate the suggestion and will include a comparison result in the final version of our paper. It is worth noting that MEIL-NeRF [1] is designed for novel synthesis tasks involving static scene stream data, whereas our approach addresses the efficient adaptation of NeRFs to dynamic real-world scene changes using minimal new images. Furthermore, our method emphasizes preserving unaltered areas, adapting to new scenes, and resolving conflicts due to scene dynamics through continual learning, which are not addressed in [1].
>
> **W2: Compare against a Generalizable NeRF.**
>
> To address this concern, we conduct comparative analyses with a generalizable NeRF called IBRNet, as detailed in Table R2 and Figure R2. Our tests, conducted with 10 new images, reveal unsatisfactory performance and serious artifacts in IBRNet's performance (denoted as MR1). Additionally, we conduct further testing on IBRNet, employing a more selective MR method involving manual removal of conflict data (denoted as MR1*). However, the obtained results remain unsatisfactory. These can be attributed to its heavy reliance on neighboring frames. To further evaluate IBRNet, we test it in different settings by varying the number of images (denoted as MR2, MR3). Although increasing the number of images improves IBRNet's performance, it still falls short compared to our model.
>
> **Q1: Are new experts added to the network from the previous task?**
>
> We would like to clarify that we employ only one expert throughout our process. For task-1, no experts are used. At task-2, one expert is added. For task-3 and subsequent tasks, we continue with just one expert, utilizing the original model combined with the expert from the previous task to distill knowledge into the current expert. This approach allows our method to efficiently adapt to new tasks while preserving knowledge from previous tasks without the need to add multiple experts.
>
> In our current exploration, this strategy is sufficient to achieve reasonable performance (see Table 1-3 in the main paper) and outperform baseline methods without increasing model complexities. We believe that adding one expert at a time may further boost performance, albeit at the cost of increased model complexity. We will study the trade-off between model complexities (adaptor numbers) and performance in the final version of our paper.
>
> **Q2: Is it necessary for the captured images in the later tasks to be local?**
>
> Yes, we opt for a local view approach in our training and evaluation for several reasons:
>
> 1) Changes within a scene typically occur in localized areas. Capturing images of altered regions helps reduce the number of images used for adapting to new changes.
>
> 2) This setting allows for a more accurate assessment of performance gains and progress, as the PSNR is calculated at the pixel level.
>
> Our approach can also adapt to images that cover global views. This is because when the captured new images contain global views, the predicted mask logit makes the adaptor primarily learn the difference, and our proposed conflict-aware knowledge distillation mechanism ensures that altered areas contained in our previously captured images do not damage the training. This enhances the robustness of our approach to scenes with global views, as these may have more conflict regions.
>
> **Q3: How the current method would scale in multiple altered regions?**
>
> To showcase the effectiveness of our method, we have developed a benchmark where 10 objects are simultaneously added and distributed across various locations within the room. The results, presented in Table R3 and Figure R3, illustrate that our method performs admirably even in challenging scenarios, achieving 32.31 and 32.15 for the old and new tasks, respectively.

---

> > ### Comment · Reviewer_fqk8 · 2023-08-13
> >
> > I have gone through the author's rebuttal and the comments from the other reviewers. The rebuttal has addressed a few concerns and I was already leaning toward an `accept' (hence keeping my score).
> >
> > Thank you for adding the comparison against IBRNet, and I am quite pleasantly surprised with its performance in this setting.
> > - I am a little confused about the white patch in Fig. R2 (in the case of IBRNet). Could the authors clarify potential reasons for the same?
> > - Could the authors also clarify how the replay data is selected in the case of MR1* and are MR2 and MR3 just more images added on top of MR1* (more specifically I would like to clarify if MR2 and 3 are selective?)
> > - Given the quite reasonable performance of IBRNet does this mean more improved methods like NeuRay, GPNR, GNT [1, 2, 3] could do even better? If it's possible I would highly appreciate one extra comparison given that there were several follow-up works after IBRNet. Nevertheless, I am still happy with the extra IBRNet experiment provided and would like to see it included in the final version of this paper.
> >
> > 1. Neural Rays for Occlusion-aware Image-based Rendering
> > 2. Generalizable Patch-Based Neural Rendering
> > 3. Is Attention All That NeRF Needs?
> >
> > - A clarification: (On average) What is the number of training images used to train the model, the number of additional images included in every task (is it independent of the operation?), and the number of conflict images removed in the case of replay?

---

> > > ### Author Response · Authors · 2023-08-15
> > >
> > > Dear Reviewer fqk8,
> > >
> > > Thank you for your insightful feedback and valuable questions. We deeply appreciate your recognition. Please find our responses to specific questions below.
> > >
> > > **Q1: Potential reasons for the white patch in Fig. R2.**
> > >
> > > Thanks for the observation. We examine the source images used in predicting Figure R2 and discover that the region with the white patch in the target frame doesn't exist in the selected source images. Thus, this issue occurs when IBRNet projects a point from the target view to the source views, causing it to be in an invalid region (e.g., outside the image domain or behind the camera plane). If the point fails to locate a corresponding position in all source views, it is labeled as an invalid pixel, and IBRNet sets its value to a default value (white in this case). In summary, this incorrect projection results in an erroneous final predicted image. This effect further verifies that IBRNet is sensitive to the number of neighboring frames.
> > >
> > > **Q2: How the replay data is selected in the case of MR1\*? Are MR2 and MR3 selective?**
> > >
> > > Yes. We would like to add more explanations.
> > >
> > > To avoid conflicts from previously captured images, we manually remove those images containing altered regions (e.g., the apple) from the old images, referred to as MR1*.
> > >
> > > Without additional manual selection, MR2 and MR3 incorporate additional images into MR1. We evaluate MR2 and MR3 to see if more images improve performance. This is motivated by the unsatisfactory results from our initial MR1 setup. We speculate that with more images, IBR could perform better.
> > >
> > > **Q3: Could more improved methods do even better?**
> > >
> > > We also conduct an experiment using GNT [3], a follow-up to IBRNet, and the results in the Dataset Whiteroom are detailed below. Despite exploring various MR methods, the findings consistently fall short of our expectations. The new task's performance may suffer from using only ten images. Similarly, the old task also lacks sufficient dense reference views. Additionally, one of the generalized NeRFs (i.e., IBRNet), stresses in their paper (Section: Sensitivity to source view density) that their approach "degrades reasonably as the input views become sparser." Thanks for your suggestions and we will include these comparisons and analyses in the final paper.
> > >
> > > | Algos | Old/New images | Old | New |
> > > |--------|--------|--------|--------|
> > > | GNT + MR1 | 200+ / 10 | 24.45 | 14.07 |
> > > | GNT + MR1* | 200+ / 10 | 24.13 | 13.98|
> > > | GNT + MR2 | 80 / 80 | 18.61 | 15.17 |
> > > | GNT + MR3 | 200+ / 80 | 26.94 | 15.80 |
> > > | **Ours** | **0 / 10** | **32.33** | **25.29** |
> > >
> > > **Q4: What is the number of training images, the number of additional images included in every task, and the number of conflict images removed in the case of replay?**
> > >
> > > While the numbers of old and new images are independent of the operation, the count after removing conflict images is operation-dependent. Specifically, the number of additional images included in every task is set to 10.
> > >
> > > For example, in Dataset Whitehouse, we train the original NeRF using 282 old images and then train our lightweight expert adaptor only using 10 newly captured images. For the MR1* setting which is used in DyNeRF, IBRNet, and GNT, conflicts removed during the ADD operation (e.g., the apple) reduce the old images to 225. Besides, the DELETE operation (e.g., the sofa) decreases the number of old images from 282 to 217.

---

> > > > ### Comment · Reviewer_fqk8 · 2023-08-15
> > > >
> > > > Thank you for your prompt response!

---

### Official Review · Reviewer_QJef · 2023-07-16

**Soundness:** 3 good
**Presentation:** 3 good
**Contribution:** 2 fair
**Rating:** 5
**Confidence:** 4

**Summary:**

This work aims to tackle the challenge of efficiently adapting NeRFs to real-world scene changes in a continual learning setting. To achieve this, it develops two techniques, including an expert adaptor for adapting to new changes and a conflict-aware knowledge distillation scheme for memorizing unchanged parts. Experiments on self-collected datasets validate the superiority of the proposed method over other potential solutions for the continual learning of NeRF.

**Strengths:**

1. As the first work targeting the continual learning of NeRF, this work could provide insights and references for the community;

2. The proposed method can achieve notable improvements over other potential solutions for the continual learning of NeRF.

**Weaknesses:**

I have the following concerns about this work:

1. The major concern is the limited technical contributions and novelty of the proposed framework. In particular, parameter-efficient tuning methods such as [1][2][3], which adapt parts of the pretrained model or newly added lightweight modules, have been widely adopted in the literature of continual learning and the proposed expert adaptor is an intuitive instantiation of these methods in the context of NeRF. And the conflict-aware knowledge distillation is more like a fine-grained memory replay thanks to the nature of 3D reconstruction tasks.

[1] "Dualprompt: Complementary prompting for rehearsal-free continual learning", Z. Wang et al., ECCV 2022.

[2] "Learning to prompt for continual learning", Z. Wang et al., CVPR 2022.

[3] "UFO: unified feature optimization", T. Xi et al., ECCV 2022.

2. It is not clear whether the proposed method can be generalized to other commonly adopted NeRF representations. For example, for voxel-based NeRFs like DVGO, the MLP part is small while most scene features are encoded in voxel embeddings. The authors are expected to discuss whether the proposed expert adaptor could also work in such scenarios.

3. What are the differences among the old tasks? If they are all unaltered regions, is the only distinction the changed camera poses? In addition, experiments on more diverse scenes, containing diverse objects and environments, are highly desirable.

4. One potential baseline is generalizable NeRF variants (e.g., IBRNet, MVSNeRF, and NeuRay), which can instantly render a new scene given a set of source views. The identified catastrophic forgetting issue can be considerably mitigated thanks to their cross-scene generalization capability even without any fine-tuning. The authors are expected to benchmark with this baseline.

**Questions:**

I have listed my questions in the weakness section.

**Limitations:**

This work targets continual learning of NeRFs, which boosts the data efficiency of NeRF training and thus does not suffer obvious negative societal impact.

---

> ### Author Rebuttal · Authors · 2023-08-09
>
> Dear Reviewer QJef,
>
> Thank you for appreciating our approach. We will address your comments below.
>
> **W1: Limited contribution and novelty, compared to [1,2,3].**
>
> 1) Existing methods in continual learning primarily deal with image classification tasks that involve adding new classes incrementally. In such tasks, there is usually no clear conflict between the old and new tasks, and therefore, the main focus is on adapting to new tasks and avoiding forgetting. In contrast, our work focuses on studying this problem within the context of NeRF, where a scene's neural radiance field is continuously evolving. In addition to overcoming the forgetting issue and adapting to new content, our task requires the model to effectively handle conflicting supervisory signals that may arise due to the overlap of old and new scene areas.
>
> 2) Our approach differs from methods like [1][2]. While these methods incorporate new trainable prompts for adapting to new tasks at the input level, we introduce new architecture designs that expand the network's capability to adapt to scene changes. Additionally, we have dedicated designs in place to handle conflicting supervisory signals. It is worth noting that the design of DyNeRF, in which we compared our approach, is similar to [1][2] in using trainable embeddings known as latent codes to capture time-specific information in the input. Our adaptor design demonstrates superiority over DyNeRF with or without Memory Replay (MR), as evidenced by the results shown in Table 2 and Figure 4 in the main paper, as well as Table 4 in the supplementary file.
>
> 3) In comparison to the current MR method, our approach effectively addresses conflicts crucial to our problem that is overlooked by existing methods. By mitigating the adverse effects caused by conflicting supervisory signals, our design significantly improves performance. This is evident from the comparison in Table 2, where executing the DELETE operation in the old task yields a performance of 34.77, compared to 17.21 in the pure data MR scenario.
>
> In summary, we believe that our proposed framework offers novel contributions to the field of NeRF, effectively addressing the challenges of adapting to scene changes and handling conflicting supervisory signals.
>
> **W2: Generalized to voxel-based NeRF (DVGO).**
>
> In response to the concern about the generalizability of our proposed method to other commonly adopted NeRF representations, we are working on extending our approach to include voxel-based NeRFs like DVGO to demonstrate its adaptability.
>
> In theory, it may also experience the forgetting phenomenon, as it first learns the old tasks and subsequently learns the new tasks with a different camera pose distribution. In such a scenario, our method could potentially be helpful in mitigating the negative impact. However, we regretfully cannot present the experimental results at this moment due to the considerable workload and time constraints for rebuttal. Nevertheless, we will promptly update and provide the experimental results during the discussion period.
>
> Besides, further experiments on diverse datasets in Table R1 highlight the general applicability of our proposed algorithm by demonstrating its robust performance in various environments.
>
> **W3: What are the differences among the old tasks? Contain more diverse scenes?**
>
> 1) We aim to clarify our approach for evaluating the old task in the following manner: Initially, we design a camera pose set to ensure comprehensive coverage of the given scene. This set assists in training an original NeRF model to represent the entire scene accurately. Following an operation or a series of operations, we can identify the unchanged areas by sampling camera poses that do not capture the alterations. To illustrate, during the execution of an ADD operation (such as adding an apple), we eliminate camera poses from the set that captures the apple. Finally, we evaluate the performance of the old task using the remaining poses from before the operation is carried out. If they are all unaltered regions, then the old task is to assess the rendering quality of the pre-trained NeRF.
>
> 2) We conduct experiments on two more challenging scenes (real-world indoor and outdoor scenes) containing various objects and environments to demonstrate the generalization ability of our algorithm. For the indoor scene, we capture images using a camera with LiDAR (specifically, an iPhone 14 Pro) and employ the Record3D App with ARKit to ensure accurate camera poses. On the other hand, the outdoor scene is captured using a DJI drone, focusing on a large-scale subject (the audience seats on a basketball court), with COLMAP utilized to calculate the camera poses. The results of these experiments are presented in Table R1 and Figure R1. We observe that Fine-tuning (FT) exhibits severe forgetting in old tasks, while Memory Replay (MR) underperforms in new tasks, particularly in qualitative results. These findings effectively showcase the robust performance of our methods across varying environments, further demonstrating the versatility and adaptability of our proposed algorithm.
>
> **W4: Compare with generalizable NeRF.**
>
> To address this concern, we conduct comparative analyses with a generalizable NeRF called IBRNet, as detailed in Table R2 and Figure R2. Our tests, conducted with 10 new images, reveal unsatisfactory performance and serious artifacts in IBRNet's performance (denoted as MR1). Additionally, we conduct further testing on IBRNet, employing a more selective MR method involving manual removal of conflict data (denoted as MR1*). However, the obtained results remain unsatisfactory. These can be attributed to its heavy reliance on neighboring frames. To further evaluate IBRNet, we test it in different settings by varying the number of images (denoted as MR2, MR3). Although increasing the number of images improves IBRNet's performance, it still falls short compared to our model.

---

> > ### Comment · Reviewer_QJef · 2023-08-15
> > **Reviewer response**
> >
> > Thank the authors for their efforts in providing the rebuttal. Most of my concerns are properly or partially answered. Although my concerns about the novelty and the generality of the proposed method are still there, given that this is the first work targeting the continuous learning setting of NeRFs, I tend to increase my score to 5 and will discuss it with other reviewers to further adjust my final score.

---

> > > ### Author Response · Authors · 2023-08-21
> > >
> > > **W2 (supplement): Generalized to voxel-based NeRF (DVGO).**
> > >
> > > As previously noted during our initial response, given the substantial workload and limited time for rebuttals, we are now presenting the experimental results regarding the extension of our approach on DVGO. This may address your valid concerns about the generalizability of our method. We sincerely appreciate your understanding and patience.
> > >
> > > DVGO employs a small MLP for RGB prediction and two distinct voxel representations for encoding density and feature information. Consequently, in the FT baseline, both the two voxels and the MLP undergo fine-tuning. In the MR baseline, the same components are fine-tuned, with the integration of old and new data following the setting outlined in our main paper. Regarding our approach, we deploy two forms of lightweight expert adaptors:
> > >
> > > **Case 1: Ours w/ MLP-based Expert**: We integrate an MLP-based expert adaptor with the original MLP to capture alterations in the modified region. During training, we only train the expert while maintaining the MLP and the feature voxel in an unaltered state. Since the scene's geometry - determined by the density voxel - changes with modifications to the scene, we also fine-tune the density voxel.
> > >
> > > **Case 2: Ours w/ MLP-based + voxel-based Expert**: In addition to the MLP-based expert adaptor, we also introduce a voxel-based expert adaptor associated with the density voxel, since the design of the expert is motivated by the insight that scene alterations are often localized, where most information learned from previous scene is expected to be kept. Specifically, the output of the voxel expert is added to the output of each original density voxel. Instead of extensively fine-tuning all model parameters, which could bring potential forgetting in unaltered regions, we focus on training only the voxel-based and the MLP-based expert adaptors, preserving the original density voxel and MLP in an unchanged state.
> > >
> > > Furthermore, in both two cases, we also utilize a conflict-aware Knowledge Distillation (KD) mechanism to maintain the consistency of density and color information in the unaltered regions. The mask logit is predicted by the MLP-based expert adaptor.
> > >
> > > Additionally, we find that the original DVGO has difficulties with novel view synthesis, showing obvious artifacts in the Dataset Whitehouse. This might be due to its dependence on the density of training images. Thus, we conduct experiments on the Dataset Rome, which offers denser training views than the Dataset Whitehouse.
> > >
> > > The table below compares the performance of our algorithm with the baselines. It also shows that FT is prone to catastrophic forgetting, while MR struggles with new tasks. The pure KD (ours w/o Expert), which fine-tunes all the voxels and the MLP, also underperforms in the tasks due to conflicting supervision signals originating from the original DVGO and the newly captured images in the modified region. Instead, our approach utilizes conflict-aware KD and an MLP-based expert which also predicts a mask logit, identifying whether the input point is altered. These designs effectively mitigate forgetting in the old task and yield impressive performance in the new task. The performance remains consistent when both MLP-based and voxel-based experts are employed. Experimental results affirm the generalizability and effectiveness of our approach.
> > >
> > > | Algorithms | Old | New |
> > > |--------|--------|--------|
> > > | MR | **27.49** | 24.18 |
> > > | FT | 23.85 | 24.76 |
> > > | Ours w/o Expert | 26.89 | 24.43 |
> > > | Ours w/ MLP-based Expert | 27.10 | 24.99 |
> > > | Ours w/ MLP-based and voxel-based Expert | 27.15 | **25.07** |
> > >
> > > Once again, we deeply value your comprehension and hope these experiments will effectively address your concerns.

---

### Author Rebuttal · Authors · 2023-08-09

Dear Reviewers and ACs:

Thank you so much for your time and efforts in assessing our paper. Hope our rebuttal has addressed your concerns. We are happy to discuss with you further if you still have other concerns. Thanks for helping improve our paper.

Best regards, Paper 1565 Authors

---

### Decision · Program_Chairs · 2023-09-21

**Decision:**

Accept (poster)

**Comment:**

Reviewers agree that the paper considers a practically significant problem and presents novel solutions. All reviewers agree for acceptance.